# Classifying Breast Cancer Subtypes Using Deep Neural Networks Based on Multi-Omics Data

**DOI:** 10.3390/genes11080888

**Published:** 2020-08-04

**Authors:** Yuqi Lin, Wen Zhang, Huanshen Cao, Gaoyang Li, Wei Du

**Affiliations:** 1Key Laboratory of Symbol Computation and Knowledge Engineering of Ministry of Education, College of Computer Science and Technology, Jilin University, Changchun 130012, China; linyq2117@163.com (Y.L.); zhangwen2017@mails.jlu.edu.cn (W.Z.); 2Center for Fundamental and Applied Microbiomics, Biodesign Institute, Arizona State University, Tempe, AZ 85287, USA; hshcao@asu.edu; 3School of Life Science and Technology, Tongji University, Shanghai 200092, China; lgyzngc@gmail.com

**Keywords:** omics data integration, breast cancer subtype, deep neural networks

## Abstract

With the high prevalence of breast cancer, it is urgent to find out the intrinsic difference between various subtypes, so as to infer the underlying mechanisms. Given the available multi-omics data, their proper integration can improve the accuracy of breast cancer subtype recognition. In this study, DeepMO, a model using deep neural networks based on multi-omics data, was employed for classifying breast cancer subtypes. Three types of omics data including mRNA data, DNA methylation data, and copy number variation (CNV) data were collected from The Cancer Genome Atlas (TCGA). After data preprocessing and feature selection, each type of omics data was input into the deep neural network, which consists of an encoding subnetwork and a classification subnetwork. The results of DeepMO based on multi-omics on binary classification are better than other methods in terms of accuracy and area under the curve (AUC). Moreover, compared with other methods using single omics data and multi-omics data, DeepMO also had a higher prediction accuracy on multi-classification. We also validated the effect of feature selection on DeepMO. Finally, we analyzed the enrichment gene ontology (GO) terms and biological pathways of these significant genes, which were discovered during the feature selection process. We believe that the proposed model is useful for multi-omics data analysis.

## 1. Introduction

Breast cancer is the most common cancer and the main cause of cancer deaths besides lung cancer in women [1]. The number of breast cancer patients is increasing year by year, and the proportion of women under 40 who have breast cancer has already reached 6.6% [2]. In 2018, there were more than 2 million new breast cancer cases worldwide [3]. At the same time, as a highly heterogeneous disease, breast cancer is composed of different biological subtypes, with different clinical, pathological, and molecular characteristics, as well as prognostic and therapeutic significance [4]. Therefore, the study of breast cancer subtypes is of great significance for precision medicine and prognosis prediction of breast cancer [5]. By understanding the molecular subtypes of breast cancer, doctors can better decide which treatment is suitable for each patient, thus saving money for the whole medical system and avoiding the side effects of unnecessary treatment [6].

The current research on breast cancer subtypes focuses mainly on the molecular typing. In 1999, molecular typing of cancer was first proposed by the National Cancer Institute (NCI) [6]. In 2000, Perou et al. first proposed the molecular typing of breast cancer and concluded that breast cancer is divided into four subtypes, namely luminal subtype, basal-like subtype, human epidermal growth subtype and normal breast-like subtype [7]. Sorlie et al. further subdivided luminal subtype into luminal A and luminal B [8]. Waks et al. classified breast cancer into three major subtypes based on the presence or absence of molecular markers for estrogen receptor (ER) and progesterone receptor (PR) and human epidermal growth factor 2 (HER2), namely ER+/PR+/HER2- (luminal A), HER2+, and triple-negative breast cancer (TNBC), which have a negative indicator in all three standard molecular markers [9]. HER2+ subtype can be further divided into two subtypes: ER+/PR+/HER2+ (luminal B) and ER-/PR-/HER2+. Tao et al. classified breast cancer into five subtypes according to immunohistochemistry (IHC) markers, including ER, PR, and HER2 [6]. These subtypes are luminal A, luminal B, HER2(+), TNBC, and unclear subtype.

In this article, breast cancer was divided into five subtypes, namely luminal A, luminal B, HER2(+), TNBC, and unclear subtype, the same classification as in a published article [10]. The detailed definition of each subtype is shown in Table 1. Luminal A is the most common breast cancer subtype, accounting for as many as 60% of all breast cancers [11]. This subtype has the highest prognosis among several breast cancer subtypes, and its 5-year local recurrence rate is much lower than other breast cancer subtypes [12]. Most patients with luminal B are elderly patients. They are similar to luminal A in that they are also sensitive to endocrine therapy. Hormone expression in patients with luminal B is reported to be lower than that of luminal A, whereas the expression and histological grade of proliferation markers are higher than those of luminal A [13]. HER2-positive breast cancer patients account for about 25%, and the prognosis is poor. Most patients with advanced HER2-positive breast cancer are most likely to metastasize to the axillary lymph nodes. In the treatment, endocrine therapy has almost no effect on it. The TNBC subtype has ER negative, PR negative, and HER2-negative [6]. Compared with other breast cancer subtypes, TNBC has rapid deterioration and metastasis. Because its three receptors are negative, targeted therapy cannot be used during the treatment of this subtype, and its prognosis is poor. Unclear subtype refers to patient samples that lack information on each of the three IHC markers.

With the explosive growth of massive biological data, the transformation of traditional biological statistical methods to computer-aided methods makes machine learning become an important part of predicting cancer prognosis [14]. If all the features in these samples are used to classify and regress, it will lead to overfitting. Feature selection or reduction, which attempts to find the subset of features that gives the model the best performance, is one of the solutions [15]. Utilizing the feature selection method can remove the obviously irrelevant and redundant gene features and improve the performance of the model. Furthermore, fewer features usually mean better interpretability and higher training speed in deep neural networks.

At present, the commonly used feature selection methods are mainly divided into the following three types: filter, wrapper, and embedded [16]. These categories are mainly based on the combination of search process of feature selection and construction process of classification models [17]. The filter methods are independent of the classifier and only rely on the intrinsic attributes of the data to select relevant features [18]. In the wrapper methods, the classification score of the features by the classifier is measured during the selection process, and the feature selection step depends on the classifier [19]. In other words, the wrapper feature selection method is to select the most fruitful feature subset for the given classifier. In the embedded methods, the step of selecting the optimal feature subset is embedded in the construction of the classifier, and the selection process can be regarded as the combined space of feature subsets and hypotheses [16]. They are completed in the same optimization process. It means that feature selection is automatically carried out during learner training. In general, when comparing complex wrappers and embedded methods to the filtering methods, the computational complexity of the former two methods is always higher, and the performance is not as good as the simple filtering method [20].

With the continuous development and improvement of high-throughput technology, there are increasingly more types of omics data obtained through high-throughput technology. Based on these omics data, there have been many studies on the classification of breast cancer subtypes. Brian D. Lehmann et al. used gene expression data to perform cluster analysis to determine the subtypes of triple-negative breast cancer [21]. Sorlie et al. achieved the classification of breast cancer subtypes by constructing a gene expression pattern based on hierarchical clustering [22]. Each type of omics data itself usually provides a list of differences associated with the disease [23]. However, the analysis of one type of omics data is limited to correlation, mainly reflecting the reaction process rather than the causal process. Multi-omics data are expected to improve the characterization of cross-molecular biological processes, and can provide more comprehensive insights into the biological systems being studied [24]. The use of multi-omics data for cancer classification has been recently suggested [25]. Multi-omics data have been used to solve different problems such as precision oncology [26], driver gene identification [27], regulatory genomics [28], and drug response prediction [29]. Artificial intelligence in cancer science includes not only classification but also diagnostics [30] as well as prediction of clinical features or identification of interactions. Most importantly, it includes research integrating multi-omics data type [31]. A further example, such as in [32], could be likewise included, as well as the one in [33]. However, there are currently few studies on the classification of breast cancer subtypes based on multi-omics data. Tao et al. used multiple kernel learning (MKL) based on multi-omics data to classify breast cancer subtypes [6]. MKL is a method widely used in multi-omics data fusion, which can improve the classification performance of original (Support Vector Machine) SVM [34]. For the classification of breast cancer subtypes, various kernels are generated and normalized using different omics data. Subsequently, after training the MKL model based on these kernels, other multi-omics data can be used to predict based on the trained model.

In this study, DeepMO, a model using deep neural networks based on multi-omics data, was employed to classify breast cancer subtypes. DeepMO contains a type-specific encoding subnetwork to learn the features of each omics type and combines features of each omics type, and a classification subnetwork is used to classify different breast cancer subtypes. In this study, the input of DeepMO contains mRNA data, DNA methylation data, and copy number variation (CNV) data, and the output of DeepMO is the predicted molecular subtypes of breast cancer. The workflow of DeepMO is illustrated in Figure 1. We compared the performances of binary classification based on multi-omics data and single omics data. Moreover, the performances of binary classification using DeepMO and MKL [6] were also compared. Then, we compared the performances of multi-classification based on multi-omics data and single omics data. Additionally, the performance of DeepMO on multi-classification was compared with some state-of-the-art data integration methods. Furthermore, we analyzed the effect of feature selection, and validated its role in classification using deep neural networks based on multi-omics data. Finally, we also analyzed the enriched gene ontology (GO) terms and biological pathways of these significant genes discovered during the feature selection process.

## 2. Materials and Methods

### 2.1. Data Sources

In this study, the data on breast cancer were collected from The Cancer Genome Atlas (TCGA) [35], which is currently a very commonly used database in the field of cancer, and contains a comprehensive range of cancer types, including various omics data and clinical data of more than 10,000 cancer patients. Among these data from TCGA, three types of omics data including mRNA, DNA methylation, and CNV data were used to compare the performance of different models. The details of the three types of omics data used are shown in Table 2.

There are 606 samples containing the three types of omics data simultaneously. Each type of omics data contains different levels of complementary information, so different aspects of breast cancer subtypes can be obtained from different perspectives. These breast cancer samples were divided into 5 subtypes, as shown in Table 3, including 277 cases of luminal A, 40 cases of luminal B, 11 cases of HER2(+), 70 cases of triple-negative breast cancer (TNBC), and 208 cases of unclear.

In addition, before performing feature selection, we normalized these data. For transcriptome data, we used the expression value from TCGA and excluded genes with missing values exceeding 200 samples. For DNA methylation data, only the regulatory relationship between gene transcription and related promoter hypermethylation or hypermethylation was considered. For CNV data, we used CNV annotation in PennCNV [36] to convert the ID of probes to gene symbols, and then merged the corresponding values according to the mapping relationship between probes and genes. In this study, we represented each type of omics data as a matrix with two dimensions, where the rows represent genes and the columns represent the subtypes of the samples.

### 2.2. Feature Selection

In general, a deep learning model does not need to select features separately because it can be done through the weight of the neural network. However, due to the “large *p* small *n*” paradigm [37] in the omics data, where *p* is the number of features and *n* is the number of samples, it is not too fruitful to use the deep learning model to train the network weights on the omics data directly. Therefore, we assumed that the feature selection algorithm of omics data can further improve the deep neural network models [38]. This is so because, as we all know, the feature selection method can remove the obvious irrelevant and redundant gene features and improve the performance of the model. Furthermore, fewer features usually mean better interpretability and higher training speed in deep neural networks.

In this study, we used the chi-squared test [39] (denoted as Chi2) to select important features. The Chi2 evaluates whether a feature in two mutually exclusive classes has a statistically significant difference [40]. For each omics type, we performed the chi-squared test separately and ranked features according to their *p*-value in hypothesis testing using corresponding samples of each classification task. Then, for each omics data, we selected the top-k feature as the input of the deep neural network. In our experiments, the value of k was set to 5000, which means that 5000 features of each omics data were input into the deep neural network.

### 2.3. The Classification Model Based on Multi-Omics Data Integration

In this study, we used a deep neural network model to classify breast cancer subtypes based on multi-omics data. The model consisted of two parts, including encoding subnetworks and a classification subnetwork that is the same as that presented in [29]. For this problem, the deep neural network model consisted of three encoding subnetworks and a classification subnetwork. After each omics type learning from its encoding subnetwork, the learned features were concatenated and input into the classification subnetwork. The entire deep neural network mode was trained as a whole. The whole process of the model can be described as follows.

#### 2.3.1. Learning Features by Encoding Subnetworks

For each omics data type, the features are learned by corresponding encoding subnetwork using corresponding omics data as input. Each type of omics data has its feedforward encoding subnetwork, which has a fully connected layer with a Relu activation function, to map the input space to the feature space. To regularize the model and enhance the training process, the encoding subnetworks use dropout and batch normalization, respectively. The input of each encoding subnetwork is one corresponding omics data and the output is the learned features. In this study, we used three omics data types to classify breast cancer subtypes, including mRNA data, DNA methylation data, and CNV data. Each of them is a *N × M* matrix with *N* samples and *M* features. In the integration step, the learned features of the different types of omics data were concatenated for obtaining the integrated representation of multi-omics, which was further smoothed by using the *l*_2_ normalization. For example, if the output of the three encoding subnetworks are three N×K feature matrices with *N* samples and *K* selected features, the output of late integration is a N×3K representation matrix.

#### 2.3.2. Cancer Subtype Classification by Classification Subnetwork

The learned features are used as the classification subnetwork to predict subtypes of breast cancer, using the concatenated presentation as input. In order to classify the subtypes of different breast cancer, the classification model utilizes a classification layer with dropout and weight decay for regularization. For binary classification, the classification subnetwork uses the Sigmoid activation function and the cost function is Binary CrossEntropy (BCE) loss. For multi-classification, we utilized Softmax regression and CrossEntropy loss instead.

### 2.4. Performance Measurements

In this study, we verified the performance of DeepMO on both binary classification and multi-classification. For binary classification, we used accuracy (Acc) and area under the curve (AUC) as performance measurements. The data of binary classification have two types of samples, positive and negative, which are represented by $P=\{P_{1}^{},P_{2}^{},\ldots ,P_{N}^{}\}$ and $N=\{N_{1}^{},N_{2}^{},\ldots ,N_{M}^{}\}$, respectively. Among them, the number of positive samples is N and the number of negative samples is M. To make some concepts easier to understand, some terms are defined as follows. The number of the correctly predicted positive samples is denoted as TP (true positives), and the number of the rest of the positive samples is denoted as FP (false positives). Similarly, we can define TN (true negatives) as the number of the correctly predicted negative samples and FN (false negatives) as the number of the negative samples. On this basis, sensitivity is defined as Equation (1), and specificity is defined as Equation (2). Since the number of different breast cancer subtype samples is imbalanced and the neural network is sensitive to the imbalance of data, the accuracy of binary classification is calculated as Equation (3). AUC is defined as the area under the ROC curve and it is no more than 1. The closer AUC is to 1, the better the performance of classification. For multi-classification, we used accuracy as the performance measurement and it can be calculated as Equation (4), where Nr represents the number of samples that are correctly predicted among all subtypes and Nt refers to the total number of all samples.
(1)$$Sen_{b}^{}=\frac{TP}{TP+FN}$$
(2)$$Spe_{b}^{}=\frac{TN}{TN+FP}$$
(3)$$Acc_{b}^{}=\frac{Sen_{b}^{}+Spe_{b}^{}}{2}$$
(4)$$Acc_{m}^{}=\frac{Nr}{Nt}$$

## 3. Results

In our experiments, we first measured the performance of binary classification of any two breast cancer subtypes. Then, we compared the performances of multi-classification based on multi-omics data and single omics data. Additionally, the performance of DeepMO on multi-classification was compared with some state-of-the-art data integration methods. Furthermore, we analyzed the effect of feature selection, and validated its role in classification using deep neural networks based on multi-omics data. Finally, we also analyzed the enrichment gene ontology (GO) terms and biological pathways of these significant genes discovered during the feature selection process. The code is available at https://github.com/linyq2117/DeepMO.

### 3.1. The Performance of Binary Classification

To investigate the performance of DeepMO on binary classification, we compared the results by using single omics data and combining three omics data based on DeepMO to classify any two subtypes of breast cancer, including (1) luminal A versus luminal B, (2) luminal A versus HER2(+), (3) luminal A versus TNBC, (4) luminal A versus unclear, (5) luminal B versus HER2(+), (6) luminal B versus TNBC, (7) luminal B versus unclear, (8) HER2(+) versus TNBC, (9) HER2(+) versus unclear, and (10) TNBC versus unclear. Specifically, the mean accuracy and AUC of 5-fold cross-validation were used as the measurements. These accuracy and AUC values on any two subtypes of breast cancer by using different types of omics data are shown in Table 4 and Table 5. From these tables, we can see that using multi-omics data can get the best accuracy and AUCs in the classification of most two subtypes of breast cancer.

In addition, to exclude the possibility of overfitting, we selected some newly annotated samples from TCGA as independent test data. The numbers of distinct breast cancer subtypes in test dataset are shown in Table 6. We trained our models on original data and tested them on independent test data. The accuracy and AUC on test data using DeepMO are shown in Table 7. We can observe that results on independent test data are similar to those on cross-validation, and it indicates that our models do not overfit.

In this study, we compared the performance of binary classification by using MKL and DeepMO. As above, accuracy and AUC were used as the classification performance measures. These accuracy and AUC values on any two subtypes of breast cancer by using MKL and DeepMO are shown in Figure 2 and Figure 3. As can be seen from these figures, DeepMO outperformed MKL in all cases.

### 3.2. The Performance of Multi-Classification

To better evaluate the performance of the proposed model, we used the model to predict breast cancer subtypes based on multi-classification. First, we compared the accuracy of multi-classification between DeepMO using single omics data and multiple omics data. The mean accuracy of 5-fold cross-validation on all subtypes of breast cancer by using single omics data and combining three omics data is shown in Figure 4. From the figure, we can conclude that DeepMO was superior to single omics data on multi-classification.

Due to the imbalance of class, which had an effect on the neural network, we took some measures to cope with it. In this study, we used both the undersampling and oversampling methods to reduce the effect of imbalanced data. We selected samples by the weight of each subtype and the weight was reciprocal to the number of each subtype. Therefore, the subtype with smaller samples had more probability to be selected. This may make our model applicable to imbalanced data. To evaluate this method, we removed all samples of one subtype each time, and performed multi-classification again. The results are shown in Figure 5. The accuracy of DeepMO is relatively stable and it indicates that imbalanced data have no dramatic impact on our model.

To further evaluate the performance of DeepMO on multi-classification, we selected some state-of-the-art methods of omics data integration, including the logistic regression model/multinational model with Elastic Net (EN) regularization [41] and Random Forest (RF) [42] in the concatenation and ensemble frameworks apart from MKL [6]. They were denoted as ConcateEN, ConcateRF, EnsembleEN, and EnsembleRF, respectively. The accuracy of multi-classification using different methods is shown in Figure 6. We can observe that DeepMO outperformed other methods in multi-classification.

### 3.3. Analysis of the Effect of Feature Selection

We utilized the chi-squared test to select the top 5000 important features before integrating multi-omics data using deep neural networks for reducing features and increasing training speed. The top 5000 features of three omics data types for multi-classification are shown in the Appendix A. Furthermore, we assumed that the feature selection algorithms can further improve the deep neural network model. To better evaluate the ability to classify each subtype, we compared the results with and without feature selection both on binary and multi-classification. The results of multi-omics data integration using deep neural networks with and without feature selection are shown in Figure 7 and Figure 8. From Figure 7, we can conclude that using feature selection can improve the accuracy on binary classification. From Figure 8, it is clear that when using feature selection, the AUC on binary classification can also be improved.

Additionally, our experiments indicated that feature selection can improve accuracy on multi-classification. The accuracy without and with feature selection on multi-classification is 0.771 and 0.782, respectively.

### 3.4. Analysis of Selected Genes

#### 3.4.1. Cluster and Biological Analysis of Selected Genes

To better analyze the important genes selected by the feature selection method, we displayed a heatmap depicted by the top 30 genes selected by feature selection on different omics data. For each omics data, the top 30 genes were ranked by *p*-values of the chi-squared test in increasing order. These heatmaps in Figure 9, Figure 10 and Figure 11 show the important genes in three types of omics data for multi-classification. It can be observed that the subtypes of breast cancer can be more clearly distinguished in mRNA and DNA methylation data. Moreover, we can observe that some genes appeared in several omics data, such as *CDK12*, *GRB7*, *ORMDL3*, *PSMD3*, *STAC2*, and *STARD3*. These genes may be significant in classifying breast cancer subtypes.

Some significant genes have been reported in related literature. ER coactivator MED1 was a novel crosstalk point for the HER2 and ERα pathways. Its expression was positively correlated to HER2 status of tumors from tissue microarray analysis of human breast cancers [43]. XBP1 was activated in TNBC and played a key role in the tumorigenicity and progression in TNBC [44]. Through the high protein expression of the known marker GRB7, HER2-positive tumors were clearly distinguished from other tumors. It was a positive marker of HER2-positive tumors [45]. STARD3 was related to the growth and survival of cancer cells amplified by HER2, and has no effect on other cancer cells [46].

#### 3.4.2. Enrichment of Selected Genes

To further understand the differences between breast cancer subtypes, we conducted an enrichment analysis of the differential genes screened from the omics data. We selected the top 1000 genes by chi-squared test on each omics data. Then, the whole set of human genes was employed as the background among the union of selected genes from three omics data using Metascape [47] against the GO and Kyoto Encyclopedia of Genes and Genomes (KEGG) Pathway databases to understand the differences in breast cancer subtypes. Metascape is a free gene annotation and analysis resource that can help biologists understand one or more gene lists. The results of KEGG pathways Enrichment are shown in Figure 12. We can see that the most significantly enriched pathway was Cell cycle. In addition, the most significantly enriched biological processes (BPs), cellular components (CCs), and molecular functions (MFs) were cell adhesion, chromosomal region, and calcium ion binding (see Appendix A).

To further analyze the relationship between GO terms of biological processes (GO_BP), we selected a rich set of terms and presented them as a network graph, where terms with similarity >0.3 were connected by edges. We chose the term with the best *p*-value from each of the 20 clusters. Each cluster did not exceed 15 terms, and the total did not exceed 250 terms. The network was visualized using Metascape, where each node represents an enriched term and is colored first by its cluster ID (Figure 13a) and then by its *p*-value (Figure 13b). From Figure 13, we can see that the most significant clusters are related to cell cycle and cell morphogenesis. Cell cycle is a series of events that lead to cell division, and the cell cycle helps to show the lowest level of cell growth. Cell morphogenesis is the basis of many aspects of cell function during development, including cell division and differentiation. Therefore, cell cycle and cell morphogenesis reflect differences in cell growth between different breast cancer subtypes.

## 4. Discussion

In this study, we utilized DeepMO, a method using deep neural networks and multi-omics late integration to classify breast cancer subtypes. First, we used the chi-squared test to select features that are closely related to labels. Afterward, we validated its performance both on binary classification and multi-classification. For binary classification, we compared accuracy and AUC obtained by DeepMO using single omics data and multiple omics data. We also compared DeepMO with MKL. The results showed that our proposed model based on multi-omics data performed best in most cases. For multi-classification, we first compared DeepMO using multi-omics data and single omics data and found that using multi-omics data can obtain better accuracy. We also compared DeepMO with other data integration methods, including ConcateEN, ConcateRF, EnsembleEN, EnsembleRF, and MKL. The compared results indicated that DeepMO was superior to all these methods. Furthermore, we compared the performance of DeepMO with feature selection and without feature selection. The results indicated that feature selection can improve DeepMO. Subsequently, we plotted the heatmap of some important genes, which visually showed that DNA methylation and mRNA data can distinguish breast cancer subtypes more clearly. Finally, we analyzed pathways and gene ontology, including KEGG, GO_BP, GO_CC, and GO_MF, and found most significant terms and similarity among them.

Feature selection is an effective approach to removing irrelevant and redundant gene features and reducing the dimension. In our model, although encoding networks can learn important features from the original gene features and achieve the reduction of features, there are still too many features inputting into encoding networks, which makes it difficult and slow to train the network weights. In addition to considering the complexity of the algorithm, we assumed that the deep neural network models may be further improved by feature selection algorithms. The results in Section 3.3 indicated the positive effects of feature selection on our model and proved our assumption well.

As we all know, the hyperparameters in deep neural networks, such as the number of nodes in the hidden layers, mini-batch size, learning rates, weight decay, the number of epochs, and dropout rate, have an important impact on the performance of the neural network. However, in this study, we only used the default or moderate values and did not adjust the hyperparameters. Therefore, the result may be better after tuning hyperparameters and selecting better values. Additionally, the imbalance of data had an effect on the neural network. In this study, we used both the undersampling and oversampling methods to reduce the effect of imbalanced data. We selected samples by the weight of each subtype and the weight was reciprocal to the number of each subtype. Therefore, the subtype with smaller samples had more probability to be selected. This can solve data imbalance to some extent.

## 5. Conclusions

The purpose of this study was to classify breast cancer subtypes by using deep neural networks based on multi-omics data from TCGA. The classification results show that, by using the proposed model, integrating multi-omics datasets can improve the performance as compared to using single omics data for classifying breast cancer subtypes. Moreover, the proposed model is superior to other state-of-the-art methods on binary classification and multi-classification. At the same time, through the analysis of important genes and pathways, we tried to find some biological explanations for the differences between breast cancer subtypes, and provide guidance for exploring the biological models of breast cancer subtypes. We believe that the proposed model is useful for multi-omics data analysis.

## Figures and Tables

**Figure 1 genes-11-00888-f001:**
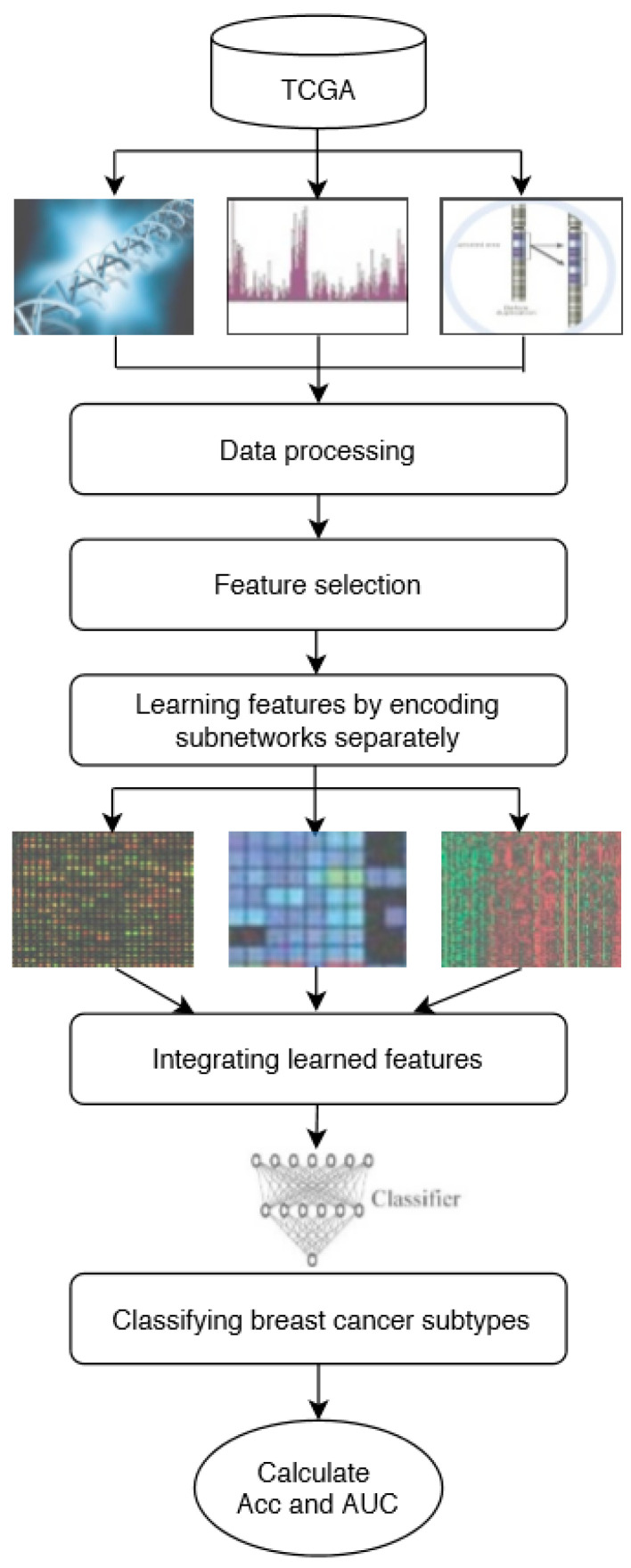
The workflow of DeepMO.

**Figure 2 genes-11-00888-f002:**
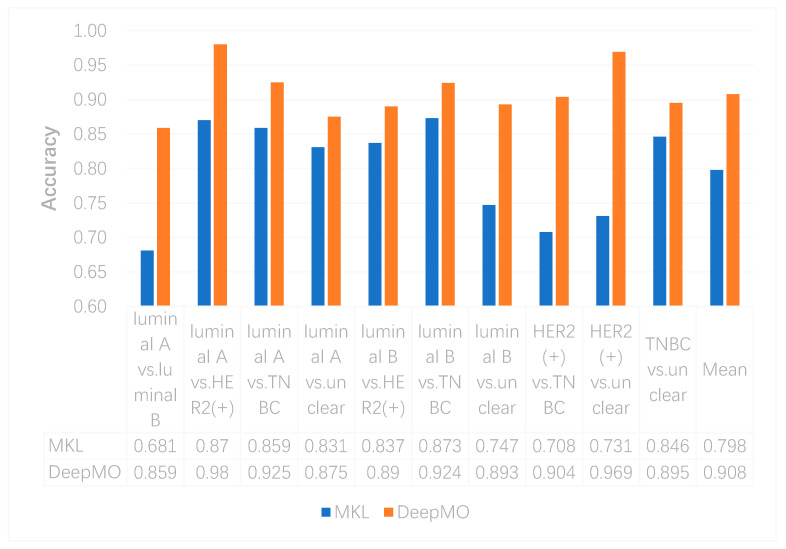
The accuracy by using multiple kernel learning (MKL) and DeepMO on binary classification.

**Figure 3 genes-11-00888-f003:**
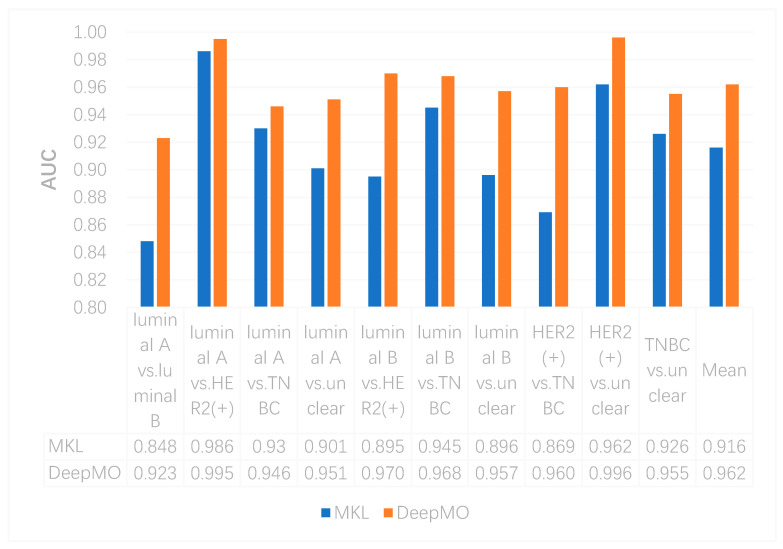
The AUCs by using MKL and DeepMO on binary classification.

**Figure 4 genes-11-00888-f004:**
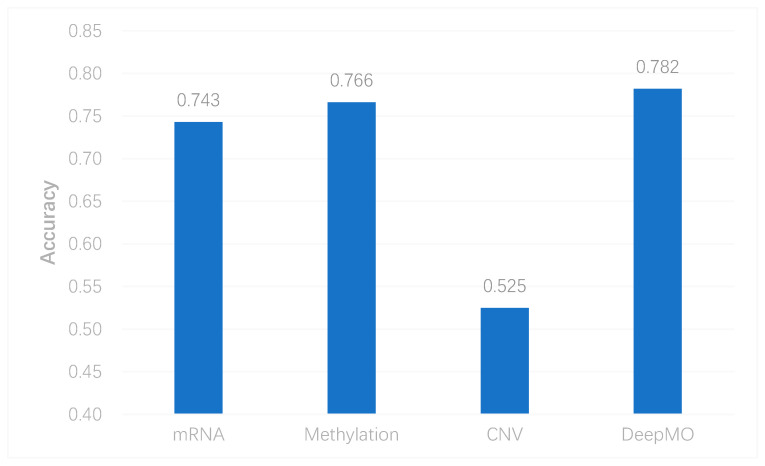
The accuracy by using single omics data and multi-omics data.

**Figure 5 genes-11-00888-f005:**
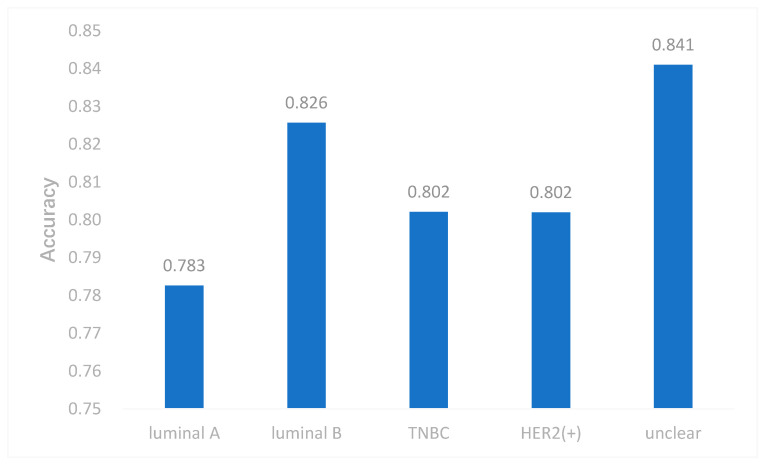
The accuracy of DeepMO when removing one subtype each time.

**Figure 6 genes-11-00888-f006:**
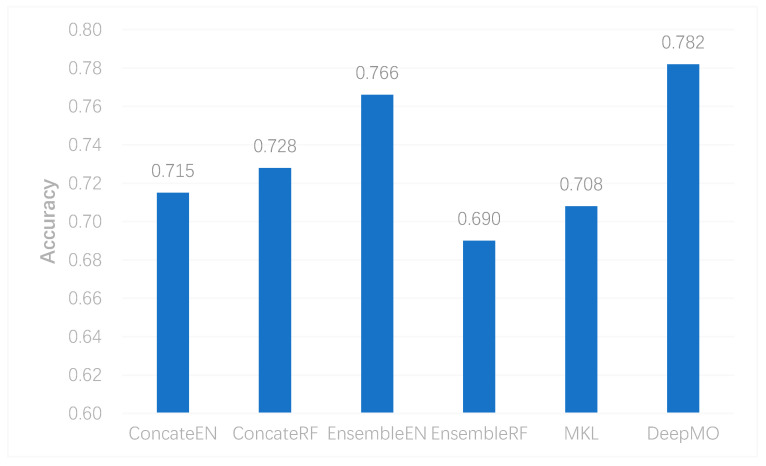
The accuracy of multi-classification using different methods.

**Figure 7 genes-11-00888-f007:**
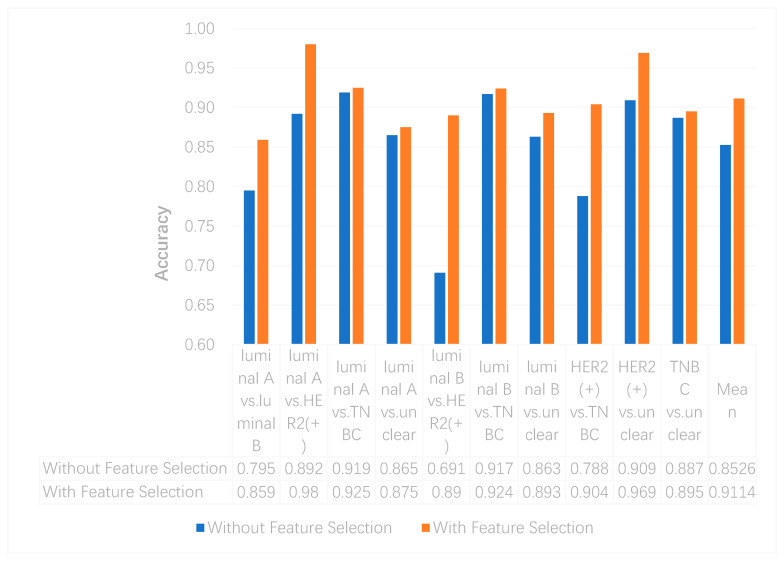
The accuracy of multi-omics data integration using deep neural networks with and without feature selection.

**Figure 8 genes-11-00888-f008:**
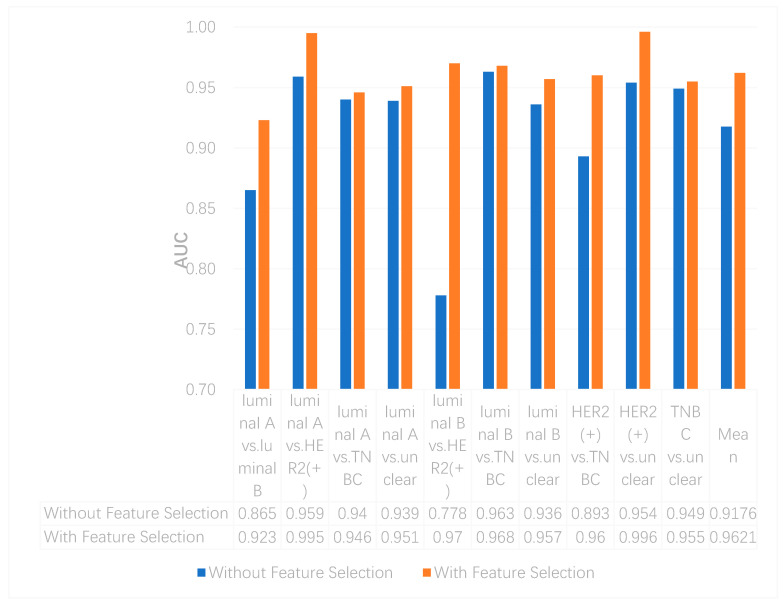
The AUCs of multi-omics data integration using deep neural networks with and without feature selection.

**Figure 9 genes-11-00888-f009:**
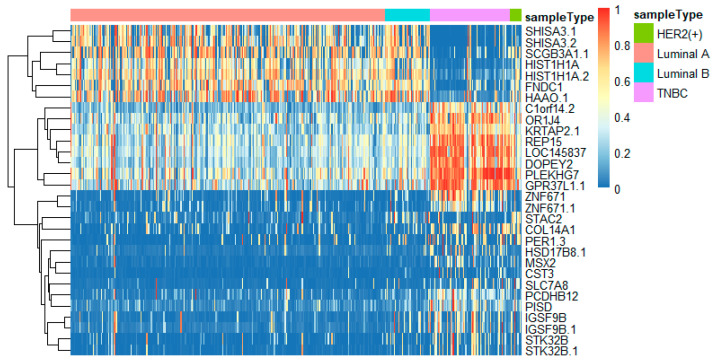
The heatmap of breast cancer subtypes on DNA methylation data.

**Figure 10 genes-11-00888-f010:**
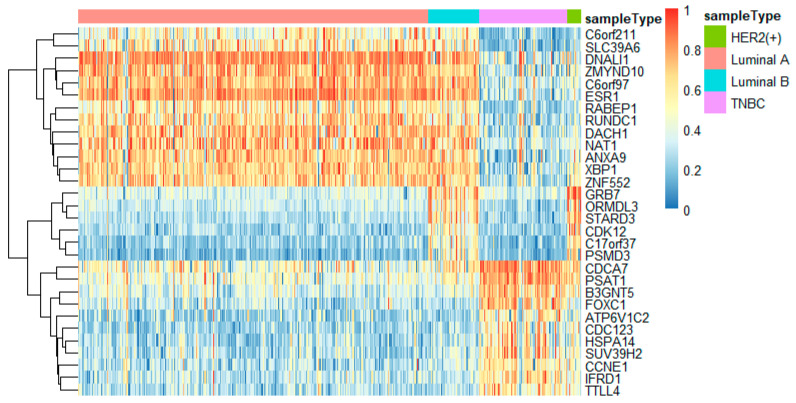
The heatmap of breast cancer subtypes on mRNA data.

**Figure 11 genes-11-00888-f011:**
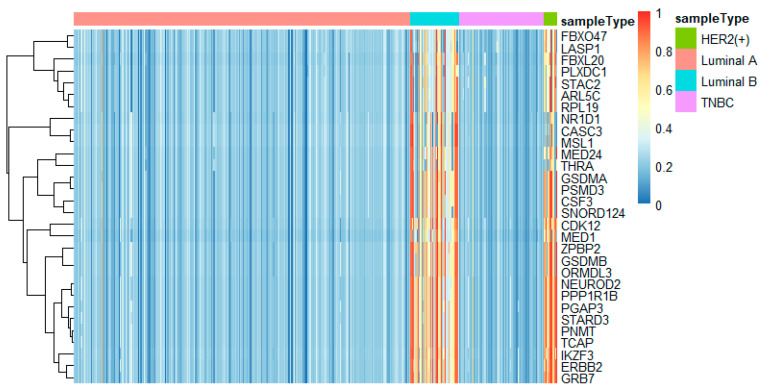
The heatmap of breast cancer subtypes on CNV data.

**Figure 12 genes-11-00888-f012:**
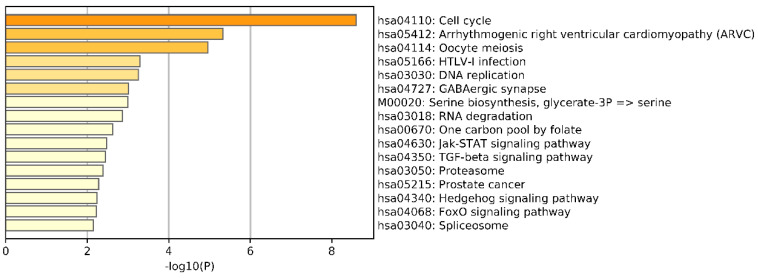
Bar graph of enriched pathways, colored by *p*-values.

**Figure 13 genes-11-00888-f013:**
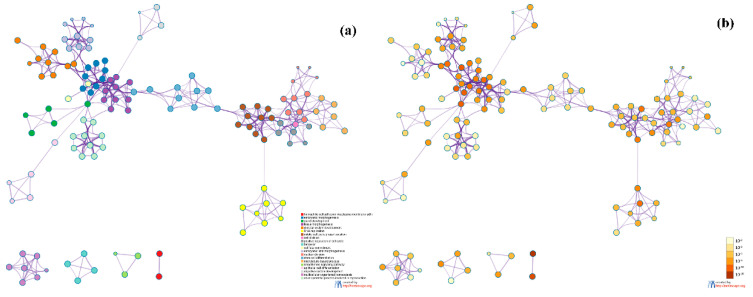
Network of enriched terms on biological processes: (**a**) colored by cluster ID, where nodes that share the same cluster ID are typically close to each other; (**b**) colored by *p*-value, where terms containing more genes tend to have a more significant *p*-value.

**Table 1 genes-11-00888-t001:** The detailed definition of breast cancer subtypes.

Breast Cancer Subtypes	IHC Markers
luminal A	ER/PR+, Her2−
luminal B	ER/PR+, Her2+
HER2(+)	ER/PR−, Her2+
TNBC	ER/PR−, Her2−
unclear	lacking

**Table 2 genes-11-00888-t002:** The summary of breast cancer data.

Data Type	Number of Samples	Number of Features	Summary
mRNA	606	13,195	RNA sequencing level 3 data
DNA methylation	606	14,285	DNA methylation 450k level 3 data
CNV	606	15,186	The Affymetrix SNP 6.0 array data with GRCH 38 (hg38) genome data

**Table 3 genes-11-00888-t003:** The numbers of distinct breast cancer subtypes.

Breast Cancer Subtypes	Number of Samples
luminal A	277
luminal B	40
HER2(+)	11
TNBC	70
unclear	208

**Table 4 genes-11-00888-t004:** The accuracy of any two breast cancer subtypes with different types of omics data and multi-omics data by DeepMO. Bold numbers are the best performance of binary classification.

Breast Cancer Subtypes	mRNA	Methylation	CNV	DeepMO
luminal A vs. luminal B	0.820	0.596	0.788	**0.859**
luminal A vs. HER2(+)	0.790	0.609	0.795	**0.980**
luminal A vs. TNBC	0.867	0.906	0.815	**0.925**
luminal A vs. unclear	0.814	0.853	0.617	**0.875**
luminal B vs. HER2(+)	0.840	0.743	0.804	**0.890**
luminal B vs. TNBC	0.911	0.895	0.889	**0.924**
luminal B vs. unclear	0.801	0.746	0.763	**0.893**
HER2(+) vs. TNBC	0.788	0.635	0.840	**0.904**
HER2(+) vs. unclear	0.748	0.697	0.660	**0.969**
TNBC vs. unclear	0.826	0.852	0.693	**0.895**
Mean	0.820	0.753	0.766	**0.908**

**Table 5 genes-11-00888-t005:** The area under the curve (AUC) values of any two breast cancer subtypes with different types of omics data and multi-omics data by DeepMO. Bold numbers are the best performance of binary classification.

Breast Cancer Subtypes	mRNA	Methylation	CNV	DeepMO
luminal A vs. luminal B	**0.944**	0.790	0.903	0.923
luminal A vs. HER2(+)	**0.998**	0.992	0.989	0.995
luminal A vs. TNBC	0.935	0.940	0.897	**0.946**
luminal A vs. unclear	0.904	0.930	0.679	**0.951**
luminal B vs. HER2(+)	0.967	0.956	0.943	**0.970**
luminal B vs. TNBC	**0.973**	0.952	0.963	0.968
luminal B vs. unclear	0.938	0.930	0.883	**0.957**
HER2(+) vs. TNBC	**0.968**	0.928	0.954	0.960
HER2(+) vs. unclear	0.986	0.987	0.977	**0.996**
TNBC vs. unclear	0.920	**0.958**	0.786	0.955
Mean	0.953	0.936	0.898	**0.962**

**Table 6 genes-11-00888-t006:** The numbers of distinct breast cancer subtypes on test data.

Breast Cancer Subtypes	Number of Samples
luminal A	51
luminal B	10
HER2(+)	2
TNBC	24

**Table 7 genes-11-00888-t007:** The accuracy and AUC of any two breast cancer subtypes with multi-omics data by DeepMO on test data.

Breast Cancer Subtypes	Accuracy	AUC
luminal A vs. luminal B	0.775	0.759
luminal A vs. HER2(+)	0.962	0.999
luminal A vs. TNBC	0.936	0.977
luminal B vs. HER2(+)	0.900	0.975
luminal B vs. TNBC	0.908	0.970
HER2(+) vs. TNBC	0.852	0.950
Mean	0.889	0.938

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
