# Peer review of "Classifying Breast Cancer Subtypes Using Deep Neural Networks Based on Multi-Omics Data"

_genes, 2020, doi:10.3390/genes11080888_

Round 1
Reviewer 1 Report
The changes in the manuscript are satisfactory. It is great to see that the authors have made their code available on github. Please provide detailed usage notes, example case studies and license condition to use their code.
Author Response
Thanks for your suggestion. We have provided some usage notes, example case studies ,example data and license condition on github.
Reviewer 2 Report
The manuscript has improved in terms of readability and reproducibility. Language should be checked once more.
Author Response
Thanks for your suggestion. We have checked our language again and rewrite some sections in our manuscript.
This manuscript is a resubmission of an earlier submission. The following is a list of the peer review reports and author responses from that submission.
Round 1
Reviewer 1 Report
General:
(G1) The study describes a new classification model of breast cancer types using various omics data. The manuscript itself should be restructured for readability and clarity. Some text passages seem to be misplaced within their current sections (examples see beneath).
(G2) Furthermore, readers may be interested into their implementation/source code for reproducibility.
Specific:
(S1) The introduction well summarizes studies on gene-based classification of breast cancer while it could be also extended in terms of general aspects of ML&AI in cancer science, not only classification but also diagnostics (McKinney, S.M. et al. International evaluation of an AI system for breast cancer screening. Nature 577, 2020), as well as prediction of clinical features or identification of interactions and foremost including other data types (Machine Learning for In Silico Modeling of Tumor Growth, 2016, in Machine Learning for Health Informatics. Lecture Notes in Computer Science, vol 9605). A further example could be likewise included (Integration of Multimodal Data for Breast Cancer Classification Using a Hybrid Deep Learning Method, 2019, In Intelligent Computing Theories and Application. Lecture Notes in Computer Science, vol 11643) as well as (Identification of cancer subtypes by integrating multiple types of transcriptomics data with deep learning in breast cancer, 2019 Neurocomputing) a.o.
(S2) The use of multi-omics data for cancer classification has been recently suggested (Elham Bavafaye Haghighi et al. Hierarchical Classification of Cancers of Unknown Primary Using Multi-Omics Data, 2019 Cancer Informatics).
(S3) Subtype descriptions (paragraph “Luminal A is the most common…”) within Material and Methods section should be moved to and integrated into the Introduction section!
(S4) General feature selection description should be likewise moved to the Introduction section as subparagraph on background information on general ML aspects amongst others to be included. In this regard, all evaluating comments on methods should be moved to the Discussion if not concluded within other studies before!
(S5) Judging comments within the Results section should be moved to the Discussion section.
Author Response
Response to Reviewer 1:
Reviewer#1, Concern # 1: The study describes a new classification model of breast cancer types using various omics data. The manuscript itself should be restructured for readability and clarity. Some text passages seem to be misplaced within their current sections (examples see beneath).
Author response: We appreciated this reviewer’s positive comment!
Author action: We have made corresponding modifications according to your suggestion, and all the changes are indicated with yellow highlighting.
Reviewer#1, Concern # 2: Furthermore, readers may be interested into their implementation/source code for reproducibility.
Author response: Insightful comments from this and other reviewers greatly improved the manuscript!
Author action: We put the GitHub address of the code in Result part.
Reviewer#1, Concern # 3: The introduction well summarizes studies on gene-based classification of breast cancer while it could be also extended in terms of general aspects of ML&AI in cancer science, not only classification but also diagnostics (McKinney, S.M. et al. International evaluation of an AI system for breast cancer screening. Nature 577, 2020), as well as prediction of clinical features or identification of interactions and foremost including other data types (Machine Learning for In Silico Modeling of Tumor Growth, 2016, in Machine Learning for Health Informatics. Lecture Notes in Computer Science, vol 9605). A further example could be likewise included (Integration of Multimodal Data for Breast Cancer Classification Using a Hybrid Deep Learning Method, 2019, In Intelligent Computing Theories and Application. Lecture Notes in Computer Science, vol 11643) as well as (Identification of cancer subtypes by integrating multiple types of transcriptomics data with deep learning in breast cancer, 2019 Neurocomputing) a.o.
Author response: Thanks for the suggestion! Indeed, it will be better if the introduction is enriched.
Author action: We have added these literatures in the introduction.
Reviewer#1, Concern # 4: The use of multi-omics data for cancer classification has been recently suggested (Elham Bavafaye Haghighi et al. Hierarchical Classification of Cancers of Unknown Primary Using Multi-Omics Data, 2019 Cancer Informatics).
Author response: Thanks for the suggestion!
Author action: We added this statement to the introduction and made corresponding changes.
Reviewer#1, Concern # 5: Subtype descriptions (paragraph “Luminal A is the most common…”) within Material and Methods section should be moved to and integrated into the Introduction section!
Author response: We are sorry that we misplaced this part.
Author action: We've moved this paragraph “Luminal A is the most common…” to the Introduction. Accordingly, the table has been updated.
Reviewer#1, Concern #6: General feature selection description should be likewise moved to the Introduction section as subparagraph on background information on general ML aspects amongst others to be included. In this regard, all evaluating comments on methods should be moved to the Discussion if not concluded within other studies before!
Author response: Thanks for this insightful comment!
Author action: We have moved the feature selection description to the Introduction as a subparagraph of background information about general ML aspects. In addition, we moved our comments on the methods to the Discussion.
Reviewer#1, Concern # 7: Judging comments within the Results section should be moved to the Discussion section.
Author response: Yes, it is proper. Thanks for the suggestion!
Author action: We have avoided using judging comments in the Results section.
Reviewer 2 Report
This work presents a deep learning classifier that learns features selected from three types of omics data - RNA-seq, methylation array data, and CNV.
It is interesting that the features of interest are not actually obtained from the neural network and this is handled during the feature selection stage. Given that the classification and feature analysis blocks are separate, the focus of the article perhaps could be changed to reflect feature selection as the main theme.
Section 2.2 briefly describes feature selection. This can be expanded further on how was the chi-square test implemented to generalize across multiple omics types? Was multiple hypothesis testing considered?
It is also not clear how were the features later combined for multi-omics? It may be useful to provide top-5000 features from each omics data as supplementary data. There is not enough information provided in the feature selection step to reproduce this analysis will similar input datasets and aims.
Section 2.3.1 talks about learning features by encoding subnetworks. The network is first trained on individual omics data and then the weights from all three blocks are concatenated. It can be wroth presenting what was the optimum number of input features found for the deep network design. How is it scalable to add another omics data, if available?
The authors achieved multi-category classification in section 3.3, it was not clear what was the aim of binary classification?
The authors performed the experiments with and without feature selection. Were all of the features of an omics data used in "without feature selection" mode? Again, it appears that the feature selection step was the key to better performance of the classifier. The authors can really strengthen this section by providing more details on methodology and validations.
The range of samples in each category is 11-277. The authors claim that their method is robust even with this class imbalance, have they assessed the classification outcome without HER(+)?
HER(+) binary classifications mostly appear to have very high AUC, do the authors have other comparable test data to assess the possibility of overfitting?
Author Response
Response to Reviewer 2:
Reviewer#2, Concern # 1: It is interesting that the features of interest are not actually obtained from the neural network and this is handled during the feature selection stage. Given that the classification and feature analysis blocks are separate, the focus of the article perhaps could be changed to reflect feature selection as the main theme.
Author response: Thanks. This is a good suggestion. In this study, we assumed that the deep neural network models may be further improved by feature selection algorithms and the results validated our hypothesis. To emphasize effect of feature selection, we added more detailed analysis on feature selection in our manuscript.
Author action: We added more detailed analysis on the effect of feature selection in our manuscript.
Reviewer#2, Concern # 2:Section 2.2 briefly describes feature selection. This can be expanded further on how was the chi-square test implemented to generalize across multiple omics types? Was multiple hypothesis testing considered?
Author response: Thanks. This is needed. For each omics type, we do chi-square separately and ranked features according to their p-value in hypothesis testing and obtained top-5000 features.
Author action: We added more details on chi-square in feature selection section.
Reviewer#2, Concern # 3: It is also not clear how were the features later combined for multi-omics? It may be useful to provide top-5000 features from each omics data as supplementary data. There is not enough information provided in the feature selection step to reproduce this analysis will similar input datasets and aims.
Author response: Yes, we did not express clearly. Each omics type had their own encoding subnetwork, after learning features from encoding subnetwork, we directly concatenated the learned features of three omics data types for obtaining one multi-omics representation. Indeed, it will be useful to provide top-5000 features from each omics data as supplementary data.
Author action: We adjusted our description in the manuscript and provided top-5000 features from each omics data as supplementary data.
Reviewer#2, Concern # 4: Section 2.3.1 talks about learning features by encoding subnetworks. The network is first trained on individual omics data and then the weights from all three blocks are concatenated. It can be wroth presenting what was the optimum number of input features found for the deep network design. How is it scalable to add another omics data, if available?
Author response:Thanks, we did not express clearly. Our deep neural network model consisted three encoding subnetworks and a classification subnetwork. After each omics type learning from their encoding subnetwork, the learned features were concatenated and input into classification subnetwork. The entire deep neural network mode was trained as a whole. If there was another omics data, we just need to add another encoding subnetwork.
Author action: We adjusted our description about this section in the manuscript.
Reviewer#2, Concern # 5: The authors achieved multi-category classification in section 3.3, it was not clear what was the aim of binary classification?
Author response: Thanks. The aim of binary classification is to better evaluate the ability to classify each subtype.
Author action: We added some descriptions on this problem.
Reviewer#2, Concern # 6: The authors performed the experiments with and without feature selection. Were all of the features of an omics data used in "without feature selection" mode? Again, it appears that the feature selection step was the key to better performance of the classifier. The authors can really strengthen this section by providing more details on methodology and validations.
Author response: Thanks. In "without feature selection" mode, all of the features of an omics data were used. It is a good suggestion that providing more details on feature selection. In fact, we assumed that the deep neural network models may be further improved by feature selection algorithms in this study and the results validated our hypothesis. To emphasize effect of feature selection, we added more detailed analysis on feature selection in our manuscript.
Author action: We added some details on feature selection.
Reviewer#2, Concern # 7: The range of samples in each category is 11-277. The authors claim that their method is robust even with this class imbalance, have they assessed the classification outcome without HER(+)?
Author response: Thanks. It is a good suggestion. We did not validated robustness of our method for class imbalance.
Author action: We add extra experiment on this problem, in which we removed all samples of one subtype each time, and performed multi-classification again.
Reviewer#2, Concern # 8: HER(+) binary classifications mostly appear to have very high AUC, do the authors have other comparable test data to assess the possibility of overfitting?
Author response: We used strategies commonly used in deep learning to reduce the risk of overfitting.
It is a good suggestion to use independent test data to evaluate overfitting.
Author action: To exclude the possibility of overfitting, we selected some newly annotated samples from TCGA as independent test data.